# Visualization of the Dynamics of Photoinduced Crawling Motion of 4-(Methylamino)Azobenzene Crystals via Diffracted X-ray Tracking

**DOI:** 10.3390/ijms242417462

**Published:** 2023-12-14

**Authors:** Koichiro Saito, Kouhei Ichiyanagi, Ryo Fukaya, Rie Haruki, Shunsuke Nozawa, Daisuke Sasaki, Tatsuya Arai, Yuji C. Sasaki, Keegan McGehee, Makoto Saikawa, Minghao Gao, Zhichao Wei, Dennis Kwaria, Yasuo Norikane

**Affiliations:** 1Research Institute for Advanced Electronics and Photonics, National Institute of Advanced Industrial Science and Technology (AIST), Tsukuba 305-8565, Ibaraki, Japan; kwaria.dennis@aist.go.jp (D.K.); y-norikane@aist.go.jp (Y.N.); 2Japan Synchrotron Radiation Research Institute (JASRI), 1-1-1 Kouto, Sayo 679-5198, Hyogo, Japan; 3Institute of Materials Structure Science, High Energy Accelerator Research Organization, Tsukuba 305-0801, Ibaraki, Japan; ryo.fukaya@kek.jp (R.F.); rie.haruki@kek.jp (R.H.); noz@post.kek.jp (S.N.); 4Graduate School of Frontier Sciences, The University of Tokyo, 5-1-5 Kashiwanoha, Kashiwa 277-8561, Chiba, Japant.arai@edu.k.u-tokyo.ac.jp (T.A.); ycsasaki@edu.k.u-tokyo.ac.jp (Y.C.S.); 5Graduate School of Science and Technology, University of Tsukuba, Tsukuba 305-8571, Ibaraki, Japan; 6Faculty of Pure and Applied Sciences, University of Tsukuba, Tsukuba 305-8571, Ibaraki, Japan

**Keywords:** azobenzene, micromanipulation, diffracted X-ray tracking

## Abstract

The photoinduced crawling motion of crystals is a continuous motion that azobenzene molecular crystals exhibit under light irradiation. Such motion enables object manipulation at the microscale with a simple setup of fixed LED light sources. Transportation of nano-/micromaterials using photoinduced crawling motion has recently been reported. However, the details of the motion mechanism have not been revealed so far. Herein, we report visualization of the dynamics of fine particles in 4-(methylamino)azobenzene (4-MAAB) crystals under light irradiation via diffracted X-ray tracking (DXT). Continuously repeated melting and recrystallization of 4-MAAB crystals under light irradiation results in the flow of liquid 4-MAAB. Zinc oxide (ZnO) particles were introduced inside the 4-MAAB crystals to detect diffracted X-rays. The ZnO particles rotate with the flow of liquid 4-MAAB. By using white X-rays with a wide energy width, the rotation of each zinc oxide nanoparticle was detected as the movement of a bright spot in the X-ray diffraction pattern. It was clearly shown that the ZnO particles rotated increasingly as the irradiation light intensity increased. Furthermore, we also found anisotropy in the rotational direction of ZnO particles that occurred during the crawling motion of 4-MAAB crystals. It has become clear that the flow perpendicular to the supporting film of 4-MAAB crystals is enhanced inside the crystal during the crawling motion. DXT provides a unique means to elucidate the mechanism of photoinduced crawling motion of crystals.

## 1. Introduction

Manipulation techniques for object motion at the microscale have been actively studied over the past few decades [1,2,3,4,5,6]. Microdroplet manipulation on solid surfaces holds promise for applications in surface patterning [7,8,9,10], microreactors [11,12,13,14,15], and bioassays [16,17,18,19,20]. Their motions are driven by external optical [21,22,23,24], thermal [25,26,27,28], electrical [29,30,31,32], and magnetic [33,34,35,36] stimuli. Optical stimulation has the advantage of allowing remote and directional control of motion. Recently, it has been reported that micron-sized azobenzene crystals exhibit crawling motion on a glass surface under light irradiation [37]. Photoinduced crawling motion of 3,3′-dimethylazobenzene crystals is driven by continuous UV and visible light irradiation from two directions. In the case of 4-(methylamino)azobenzene (4-MAAB) crystals, negative phototaxis can be achieved simply by irradiating with monochromatic visible light [38]. Furthermore, we have achieved nano-/micromaterial transportation using photoinduced crawling motion of 4-MAAB crystals [39]. The ability to remotely control material transportation on an unmodified solid surface is a great advantage in micromanipulation techniques.

During photoinduced crawling motion, part of the azobenzene crystal undergoes a phase transition to liquid. Such a phase transition was reported in macrocyclic azobenzenes [40] and small crystalline molecules [41,42,43,44,45,46,47,48,49]. Crystals of 3,3′-dimethylazobenzene (DMAB) are melted by photoisomerization from the trans to the cis isomer under UV irradiation. The cis isomer in the liquid state recrystallizes via back-isomerization with heat or visible light irradiation. When light is irradiated obliquely onto a crystal on a glass substrate, the penetration depth of the light within the same crystal varies depending on the location [50]. As a result, both melting and recrystallization sites coexist within the same crystal. The non-equilibrium state of phase transition caused by continuous light irradiation is presumed to be the driving force behind the crawling motion.

In the case of 4-MAAB, the electron-donating amino group at the para position lowers the energy required for thermal back isomerization. When the trans isomer of 4-MAAB photoisomerizes to the cis isomer by monochromatic visible light, the cis isomer thermally isomerizes to the trans isomer within 1 s at room temperature (Figure 1a). Therefore, photoisomerization and thermal back isomerization are repeated continuously under the monochromatic visible light irradiation. Similar to DMAB, site-dependent heterogeneity of melting and recrystallization could be the driving force for the crawling motion of 4-MAAB crystals. However, the detailed mechanism of photoinduced crawling motion remains unclear because spectroscopic studies of the internal dynamics of 4-MAAB crystals are difficult.

Herein, we used the diffraction X-ray tracking (DXT) method to investigate the internal dynamics of the photoinduced crawling motion of 4-MAAB crystals. The DXT method, which is based on time-resolved X-ray diffraction (XRD), is a technique for analyzing the rotational motion of crystals from the movement of bright diffraction spots in their diffraction patterns [51]. It has been widely used to observe the dynamics of membrane proteins [52], supersaturated proteins [53], etc. In this study, the dynamics of zinc oxide (ZnO) particles mixed in 4-MAAB crystals (Figure 1b) was analyzed by the DXT method. Diffraction spots from ZnO particles in 4-MAAB crystals moved on the XRD pattern under light irradiation, revealing their rotational dynamics. The rotational motion of the ZnO particles increased as the intensity of the irradiating light increased. Furthermore, the anisotropy in the rotational direction of ZnO particles, which is unique to the crawling motion of 4MAAB crystals, was demonstrated. DXT analysis is an excellent technique to visualize the rotational dynamics that occur during motion manipulation of nano-/microscale objects.

## 2. Results

### 2.1. Time-Resolved XRD of 4-MAAB Crystal Containing ZnO Particles

We conducted time-resolved XRD experiments under light irradiation using a fine powder of 4-MAAB crystals, which contain ZnO particles inside. Note that this sample is more accurately described as “a two-phase polycrystalline sample of ZnO and 4-MAAB”. Here, we described the sample as “4-MAAB crystal” for simplicity. The content of the ZnO particles was at a weight ratio of 1:10 to the 4-MAAB crystals. The scanning electron microscopy image of ZnO particles is shown in Appendix A. The hydrodynamic diameter of the ZnO particles measured by dynamic light scattering was approximately 600 nm. We have previously reported that silica particles with a diameter of 3 μm can be transported by 4-MAAB crystals [39]. Therefore, 4-MAAB crystals can also transport ZnO particles with a diameter of approximately 600 nm.

A photograph of the experimental setup is shown in Appendix A. The polyimide film supporting the 4-MAAB crystals was fixed vertically. Since the incident direction of the synchrotron X-ray was horizontal, the X-ray was incident perpendicularly to the sample from the back side (Figure 1c). Blue LED light was incident on the polyimide film at an incident angle of 60° to induce photoinduced crawling motion. The peak emission wavelength of the blue LED light is 465 nm. Condenser lenses were used to irradiate the 4-MAAB crystals with sufficient intensity of light. In this experimental setup, the 4-MAAB crystals exhibit photoinduced crawling motion in a direction that opposes gravity, as we have previously reported [39]. The light intensity was adjusted in the range of approximately 0–250 mW cm^−2^. The experiment was conducted at room temperature (25 °C). In order to reduce the damage to the sample caused by X-rays, we performed XRD experiments by shifting the X-ray irradiation position by a larger distance than the beam size (200 μm) for each measurement.

The XRD pattern of 4-MAAB powder containing ZnO particles before light irradiation is shown in Figure 1d. Many bright diffraction spots were observed along the diffraction rings corresponding to (100), (002), and (101) planes of ZnO crystals. The diffraction pattern located inside the diffraction rings of ZnO crystals originates from 4-MAAB crystals. These diffraction rings were blurred because the energy width of the X-ray used in this study was wider than that of monochromatic X-ray. X-rays with a wide energy width enabled tracking of the movement of the diffraction spot. The rotation of the ZnO particles, with their rotational axis parallel to the polyimide film, corresponds to the movement of the diffraction spot in the θ direction (Figure 1c,d). On the other hand, the rotation with the axis perpendicular to the polyimide film corresponds to the movement in the χ direction.

Figure 1e and Appendix A show the typical movement of diffraction spots on the sample under light irradiation. Light was irradiated 1 s after the start of the XRD measurement. The diffraction spots from the ZnO particles moved around violently in all directions. Such movement means that the ZnO particles rotate in both the θ and χ directions under light irradiation. The movement of each diffraction spot was tracked by capturing diffraction images every 50 ms. Based on DXT analysis, we investigated how ZnO particles rotate inside 4-MAAB crystals under light irradiation.

### 2.2. DXT Analysis for Polycrystalline 4-MAAB Powder

XRD measurements were carried out under light irradiation using 4-MAAB powder containing ZnO particles inside, as shown in Figure 2a. To examine the dynamics in the steady state under light irradiation, the movement of the diffraction spots immediately after the start of light irradiation was excluded. Diffraction from the 4-MAAB crystals was excluded by masking the XRD pattern. The diffraction spots from the rotating ZnO particles appear and disappear during the XRD measurements, according to Bragg’s law. Therefore, we defined the time when each diffraction spot first appeared as the start time of its respective movement. Each rotational displacement was calculated based on the movement detected during the time interval (Δt) from the start time.

The histograms of the rotational displacements of the ZnO particles under light irradiation are shown in Figure 2b,c. The frequency in the histogram is the number of diffraction spots that appear in the XRD pattern. The central bin of the histogram represents zero displacement, that is, the number of diffraction spots that did not move during light irradiation. Therefore, the decrease in the central bin means a decrease in the number of ZnO particles that did not rotate. The rotational displacement of ZnO particles was calculated from the moving distance of the diffraction spot. As shown in Figure 1c, the rotational displacement of ZnO particles exhibited positive or negative values. Therefore, the increase in bins left and right from the center of the histogram means an increase in rotated ZnO particles.

The rotational displacements detected during the time interval (Δt) of 1000 ms are clearly larger than for 100 ms (Figure 2b,c). This result indicates that the ZnO particles continuously rotated in the θ and χ directions under light irradiation. Compared to the rotational displacements in the θ direction, the rotational displacements in the χ direction tended to show larger values. In DXT analysis, the rotational displacements in the χ direction show a larger value because the angle per pixel on the XRD pattern images is larger for χ than θ.

Next, we investigated the dependence of the rotational motion of the ZnO particles on the light intensity. Figure 2d shows the histograms of the rotational displacements in the θ and χ directions in the absence of light irradiation. The ZnO particles showed slight rotation even without light irradiation. Damage to the 4-MAAB crystals due to high-energy X-ray irradiation could be the cause of the perturbation. On the other hand, the light irradiation caused the diffraction spot to move more violently. As shown in the histogram in Figure 2e, the rotational displacements increased slightly at a light intensity of approximately 100 mW cm^−2^. At a light intensity of approximately 200 mW cm^−2^, the displacements clearly increased (Figure 2f). In particular, the histogram of displacements in the θ direction has a broadened tail.

We quantitatively evaluated the intensity of the rotational motion of the ZnO particles by calculating the mean square displacement (MSD). Figure 2g,h shows the MSD in the θ and χ directions, respectively. It was clearly demonstrated that the rotational motion became more intense as the light intensity increased. In this experiment, each 4-MAAB crystal was partially melted by light irradiation, allowing the ZnO particles to rotate due to the flow of the molten 4-MAAB. A possible reason for the light intensity dependence of MSD is that the stronger the light intensity, the more 4-MAAB melted, leading to easier rotation of ZnO particles.

Focusing on light intensities of 200 and 250 mW cm^−2^, the MSD in the θ direction tended to increase exponentially, while the χ direction tended to saturate. The rotation of the ZnO particles could be increased, specifically in the θ direction, due to the crawling motion of the 4-MAAB powder. To confirm that the crawling motion contributes to the rotation in the θ direction, experiments on samples that do not exhibit crawling motion under light irradiation are required. In addition, it is also necessary to investigate the dynamics of ZnO particles when the 4-MAAB crystals are completely melted.

### 2.3. DXT Analysis for a 4-MAABcrystalline Film

A 4-MAAB polycrystalline film containing ZnO particles, which does not exhibit photoinduced crawling motion, was deposited on a polyimide film (Figure 3a). The 4-MAAB film was sandwiched between polyimide films and had a thickness of approximately 30 μm. Since the 4-MAAB film sandwiched between polyimide films was further fixed to the holder, no crawling motion occurred. In this experiment, we also aimed to investigate the dynamics of ZnO particles when 4-MAAB crystals are completely liquefied. The melting point of 4-MAAB is 87 °C [38]. Therefore, the experiment was carried out with the sample temperature at approximately 75 °C by blowing hot nitrogen gas.

As shown in Figure 3b,c, the rotational displacements of ZnO particles in the 4-MAAB film increased with time under light irradiation. In both the θ and χ directions, there was no noticeable difference in the temporal change in the displacements between the 4-MAAB powder and the film. It was confirmed that ZnO particles exhibit rotational motion in 4-MAAB crystals even without photoinduced crawling motion. Similar to the powder, the crystalline film also showed an increase in the rotational displacements depending on the irradiation light intensity (Figure 3d–f). However, the degree of the displacements was different in the χ and θ directions. At the light intensity of 200 mW cm^−2^ (Figure 3f), while the displacements in the χ direction were significantly large, the displacements in the θ direction appeared to be smaller than in the case of powder.

The MSD values were similar at the light intensities of 0, 50, and 100 mW cm^−2^ (Figure 3g,h). Low light intensity did not sufficiently melt the 4-MAAB crystals, which prevented rotation of the ZnO particles. On the other hand, when the light intensity was sufficiently high, the MSD increased significantly, especially in the χ direction at 200 mW cm^−2^. The reason is presumed to be the complete melting of the 4-MAAB crystalline film during the XRD measurement.

Figure 4a,b are photographs of the 4-MAAB film 1 s before the start of 200 mW light irradiation and 1 s after the irradiation was stopped, respectively. After the light irradiation was stopped, the part where the crystalline film had been liquefied appeared dark due to the background showing through, indicating that aggregates of ZnO particles were floating. Appendix A shows how the 4-MAAB crystalline film melted. Figure 4c,d are the XRD patterns of the 4-MAAB film containing the ZnO particles 1 s before the start of light irradiation and 1s after the irradiation was stopped, respectively. While the diffraction rings from the 4-MAAB crystalline film disappeared by light irradiation, the diffraction spots from the ZnO particles remained. Complete melting of the 4-MAAB crystal was also confirmed by the XRD measurement. The changes in the XRD diffraction pattern are also shown in Appendix A.

In Appendix A, the diffraction spots moved significantly, especially at the moment of melting. The MSD with such extreme movements does not reflect the steady-state rotational dynamics of the ZnO particles in the completely melted 4-MAAB crystal. Therefore, DXT analysis was performed on the movement of the diffraction spots after the 4-MAAB crystal was completely melted. By excluding the movement at the moment of melting, the MSD in both the θ and χ directions decreased (Appendix A). However, we found that when 4-MAAB was completely melted, the rotation of ZnO particles was more intense than when 4-MAAB remained crystalline.

The above results demonstrate that the ZnO particles rotate in molten 4-MAAB even in the system where photoinduced crawling of 4-MAAB crystals does not occur. The driving force for the rotation could be the flow of the liquid 4-MAAB due to thermal convection and recrystallization. Therefore, there appears to be no significant difference in the rotational dynamics of the ZnO particles within the 4-MAAB powder and the crystalline films. However, focusing on the MSD in the θ direction, the increasing manners for both appear to be slightly different. At the light intensities of 150 mW cm^−2^ and 200 mW cm^−2^, an exponential increase in MSD was observed for the powder and a saturating increase in MSD for the film.

## 3. Discussion

To compare the rotational anisotropy of the ZnO particles in 4-MAAB powder and crystalline films, we calculated the ratios of the MSD in the θ direction (MSD_θ_) and the MSD in the χ direction (MSD_χ_), respectively. The value of MSD_θ_/MSD_χ_ could differ from the actual rotational anisotropy. However, comparing the MSD_θ_/MSD_χ_ between different samples is useful for finding the differences in the rotational dynamics. Figure 5a shows the light intensity dependence of the rotational anisotropy of ZnO particles in 4-MAAB powder. The rotational anisotropy in the θ direction increased at higher light intensities. On the other hand, the MSD_θ_/MSD_χ_ for the 4-MAAB crystalline film showed similar values at any light intensity (Figure 5b). Although the ZnO particles in the crystalline film rotate violently as the light intensity increases, the rotational anisotropy did not change clearly. The clear difference in rotational anisotropy between the powder and the film could be due to the photoinduced crawling motion of the 4-MAAB crystals.

We observed photoinduced crawling motion of 4-MAAB crystals containing ZnO particles supported on a polyimide film using an optical microscope. Blue LED light was irradiated from the back side of the polyimide film at an incident angle of 60°. The light intensities were 60, 100, 200, and 300 mW cm^−2^. At the light intensities of 60 and 100 mW cm^−2^, the 4-MAAB crystals hardly moved from their original positions (Appendix A). On the other hand, a clear crawling motion was observed at the light intensities of 200 and 300 mW cm^−2^ (Appendix A). The average velocities calculated from five 4-MAAB crystals were approximately 0.68 ± 0.43 and 0.72 ± 0.46 μm min^−1^ at light intensities of 200 and 300 mW cm^−2^, respectively. Since the optical system of this experiment was different from that of the DXT experiment, it is impossible to directly compare the light intensity dependence of the crawling motion. However, the result that light irradiation with intensity above a threshold is required to induce a crawling motion is significant.

At low light intensity, the ZnO particles could rotate in random directions inside the 4-MAAB powder, similar to inside the crystalline film. The MSD_θ_/MSD_χ_ corresponding to the light intensity of 100 mW cm^−2^ and 150 mW cm^−2^ were almost the same (Figure 5a). As shown in Figure 2g,h, both MSD_θ_ and MSD_χ_ at each light intensity are clearly different. Nevertheless, there was no difference in the rotational anisotropy. It is possible that there was almost no crawling motion at the light intensity of 100 mW cm^−2^ and 150 mW cm^−2^. This supports the assumption that the rotation increase in the θ direction in response to the light intensity originates from the photoinduced crawling motion.

Based on the above results, we hypothesized that the liquid 4-MAAB flowed in a direction that specifically induced rotation in the θ direction during the photoinduced crawling motion. Assuming a cylindrical particle in an incompressible fluid, the torque exerted on the particle is given by
(1)$$T=\oint_{\Gamma_{p}}\left(r-r_{p}\right)\times \left(\sigma \cdot dS\right)$$
where r is the position vector on the particle plane $\Gamma_{p}$ and $r_{p}$ is the location of the cylinder’s center of mass [54]; σ is the stress tensor given by
(2)$$\sigma =-pI+\mu \left[\nabla \upsilon +{\left(\nabla \upsilon \right)}^{T}\right]$$
where p and υ are the fluid pressure and velocity fields with the viscosity μ. Since the ZnO particles were hexagonal prisms, which have a wurtzite structure (Appendix A), we considered them as cylindrical particles. Also, since the fluid was a single-component 4-MAAB liquid, capillary stress was ignored. Depending on the flow velocity, a torque is exerted on the particles along the direction of flow.

As shown in Figure 5c,d, there are two types of flow directions that drive rotation in the θ direction. It has been reported that the velocity of photoinduced crawling motion of 4-MAAB crystals increases as the irradiation light intensity increases [38]. Since the flow parallel to the polyimide film corresponds to the direction of the crawling motion, the flow velocity could increase in that direction. This finding is expected to contribute to the enhancement of the velocity of photoinduced crawling motion.

The flow in the direction perpendicular to the polyimide film is orthogonal to the direction of the photoinduced crawling motion (Figure 5d). It seems unnatural that the flow perpendicular to the supporting film increases with the crawling motion. However, our previous report has shown that particles within the 4-MAAB crystals move perpendicular to the direction of photoinduced crawling motion [39]. Moreover, perpendicular flow does not contribute to the rotation in the χ direction, unlike parallel flow. Therefore, it is reasonable to assume that the perpendicular flow also influenced the increase in the rotational anisotropy (MSD_θ_/MSD_χ_) in the θ direction.

From the above discussion, it was suggested that the 4-MAAB crystal could have a rolling, crawling motion due to the perpendicularly circulating flow. However, while our experiments revealed the dynamics of the ZnO particles, a direct analysis of the rotational dynamics of the 4-MAAB crystals themselves was not possible. Since the grain size of the 4-MAAB powder is approximately several tens of microns, which is much larger than the ZnO particles, they were not suitable for DXT analysis. We hope that deeper insight into the mechanism of photoinduced crawling motion will be obtained by preparing 4-MAAB crystalline particles with a size suitable for DXT analysis in the future.

In this study, we successfully visualized the rotational dynamics of ZnO particles in 4-MAAB crystals under light irradiation through DXT analysis based on time-resolved X-ray diffraction. It was demonstrated that the ZnO particles rotated violently in response to the irradiated light intensity. Furthermore, it was revealed that photoinduced crawling motion of the 4-MAAB crystals induced anisotropic rotation of the ZnO particles. These results indicated the possibility that the 4-MAAB crystals themselves could rotate while crawling. DXT analysis was very useful in elucidating the mechanism of photoinduced crawling motion. In addition, DXT analysis, which enables investigation of rotational anisotropy of particles in fluids, would contribute to the development of micromanipulation techniques.

## 4. Materials and Methods

### 4.1. Sample Preparation for Time-Resolved XRD Experiments

The 4-methylaminoazobenzene (4-MAAB) was purchased from Tokyo Chemical Industry Co., Ltd. (Tokyo, Japan). The 4-MAAB was purified using silica gel column chromatography and subsequent recrystallization from hexane. ZnO particles were purchased from EM Japan Co., Ltd. (Tokyo, Japan). The morphology of the ZnO particles was observed using scanning electron microscopy (JSM6700F, JEOL Ltd., Tokyo, Japan), and the average particle size was measured using a dynamic light scattering (DLS) measurement system (zetasizer nano ZS, Malvern Instruments Ltd., Malvern, Worcestershire, UK). The wavelength of the laser used for the DLS measurement was 532 nm.

The ZnO particles were dispersed in toluene, and 4-MAAB was dissolved in the dispersion. The 4-MAAB solution in which the ZnO particles were dispersed was heated on a hot plate at 95 °C to evaporate toluene. This process produced a liquid 4-MAAB with the dispersed ZnO particles. After cooling to room temperature, 4-MAAB crystals containing the ZnO particles inside were obtained. By grinding these 4-MAAB crystals in a mortar, the fine polycrystalline powder used in this experiment was obtained. The powder sandwiched between two polyimide films was used for the XRD measurements. The polyimide film (PI-25 μm-A4, 4-1355-01) was purchased from AS ONE CORPORATION (Osaka, Japan). The polyimide film does not strongly absorb LED light with a wavelength of 465 nm, as shown in Appendix A. The transmittance spectrum of the polyimide film was measured with a v-730 spectrophotometer (JASCO Corporation, Tokyo, Japan).

A liquid film of 4-MAAB was formed by melting the 4-MAAB powder sandwiched between two polyimide films and heating it to 95 °C on a hot plate. By recrystallizing it at room temperature, a 4-MAAB polycrystalline film containing ZnO particles was obtained. The thickness of this film was made to be approximately 30 μm using a spacer. A temperature controller (Cryostream, Oxford Cryosystems, Oxford, UK) that sprays nitrogen gas was used to heat the 4-MAAB crystalline film. An infrared (IR) camera (FLIR i3, Teledyne FLIR LLC, Wilsonville, OR, USA) was used to measure the temperature of the sample.

### 4.2. DXT Measurements

DXT measurements were conducted at the High Energy Accelerator Research Organization (KEK) Photon Factory Advanced Ring (PF-AR) NW14A (Tsukuba, Japan). The white X-ray had a peak position of approximately 20 keV and a bandwidth of approximately 35% (Appendix A). Time-resolved X-ray diffraction was recorded using a 2D photon counting detector (Pilatus 100 K, Dectris, Baden-Dättwil, Switzerland). The beam size of the X-ray was set to 200 μm × 200 μm. For each measurement, 1000 diffraction images were recorded in 50 s. In other words, each diffraction image was recorded every 50 ms. A blue LED (465 nm: HLV2-22BL-3W, CCS Inc., Kyoto, Japan) was used for photo-irradiation. Light intensities were measured using a Newport 1917-R optical power meter with an 818-ST-UV photodetector. Light irradiation and DXT started 1 and 5 s after the start of the XRD measurement, respectively.

Each diffraction spot from ZnO particles was tracked by TrackPy (v0.6.1, https://doi.org/10.5281/zenodo.7670439 (accessed on 3 August 2023)) after excluding the diffraction from 4-MAAB crystals by masking using ImageJ software (v1.52p). Analysis of the trajectory of each diffraction spot was performed using a Python program (Python 3.9.12). The time and position at which each diffraction spot first appeared were defined as the start time and start position of movement, respectively. For each diffraction spot, the rotational angular displacement (χ, θ) from the start position during the time interval (Δt) was calculated. Mean square displacement (MSD) was calculated by squaring the average of the absolute values of the rotational angular displacements. The MSD curves were plotted as a function of the time interval (Δt).

## Figures and Tables

**Figure 1 ijms-24-17462-f001:**
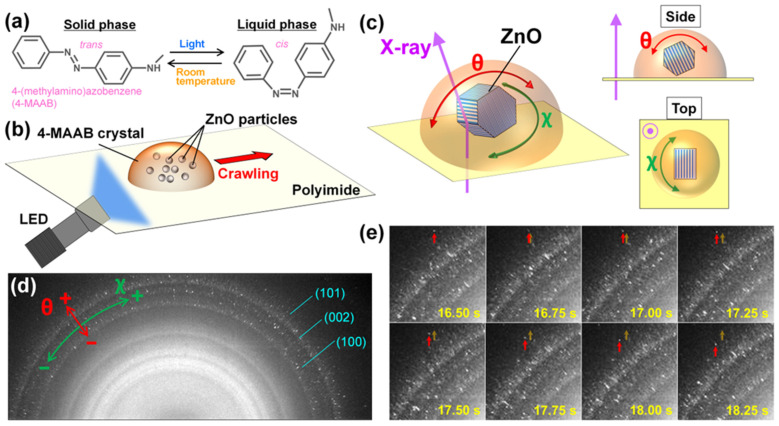
(**a**) Scheme of photoisomerization and thermal back-isomerization of 4-(methylamino)azobenzene (4-MAAB); (**b**) schematic illustration of the photoinduced crawling motion of a 4-MAAB crystal containing ZnO particles. Blue LED light was obliquely incident on the sample from the back side; (**c**) rotation directions of ZnO particles in a 4-MAAB crystal. The X-ray was incident perpendicularly to the polyimide film on which the 4-MAAB crystals were deposited. For rotations in the θ and χ directions, the X-ray incident direction and the rotation axis were perpendicular and parallel to each other, respectively; (**d**) XRD pattern of 4-MAAB powder containing ZnO particles inside. Diffraction rings with bright diffraction spots correspond to (100), (002), and (101) planes of ZnO crystal. The inner diffraction rings without diffraction spots arise from the 4-MAAB powder. The movement of the diffraction spot towards the outside of the diffraction ring corresponds to a positive rotational displacement in the θ direction. The clockwise movement of the diffraction spot on the diffraction ring corresponds to a positive rotational displacement in the χ direction; (**e**) movement of a diffraction spot captured every 250 ms under light irradiation. The light intensity was 200 mW cm^−2^. The red arrows point to the diffraction spot, and the orange arrows point to the original position.

**Figure 2 ijms-24-17462-f002:**
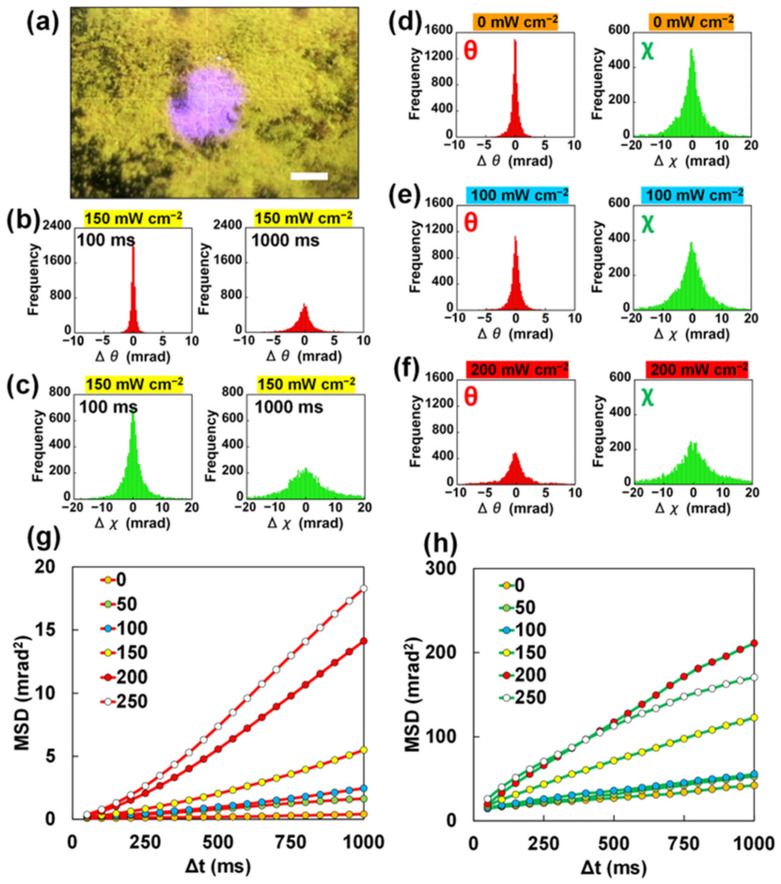
(**a**) Photograph of 4-MAAB powder containing ZnO particles (scale bar: 200 μm); (**b**,**c**) histograms of rotational displacements of ZnO particles in 4-MAAB powder in (**b**) θ direction and (**c**) χ direction at the light intensity of 150 mW cm^−2^. Displacements detected during the time interval (Δt) of 100 ms and 1000 ms are shown, respectively; (**d**–**f**) histograms of rotational displacements of ZnO particles in the 4-MAAB powder in θ direction and χ direction at the light intensity (**d**) 0, (**e**) 100, (**f**) 200 mW cm^−2^. Displacements detected during the time interval (Δt) of 1000 ms are shown; (**g**,**h**) MSD curves of (**g**) θ direction and (**h**) χ direction calculated from the rotational displacements as a function of the time interval (Δt) at the light intensities 0, 50, 100, 150, 200, and 250 mW cm^−2^.

**Figure 3 ijms-24-17462-f003:**
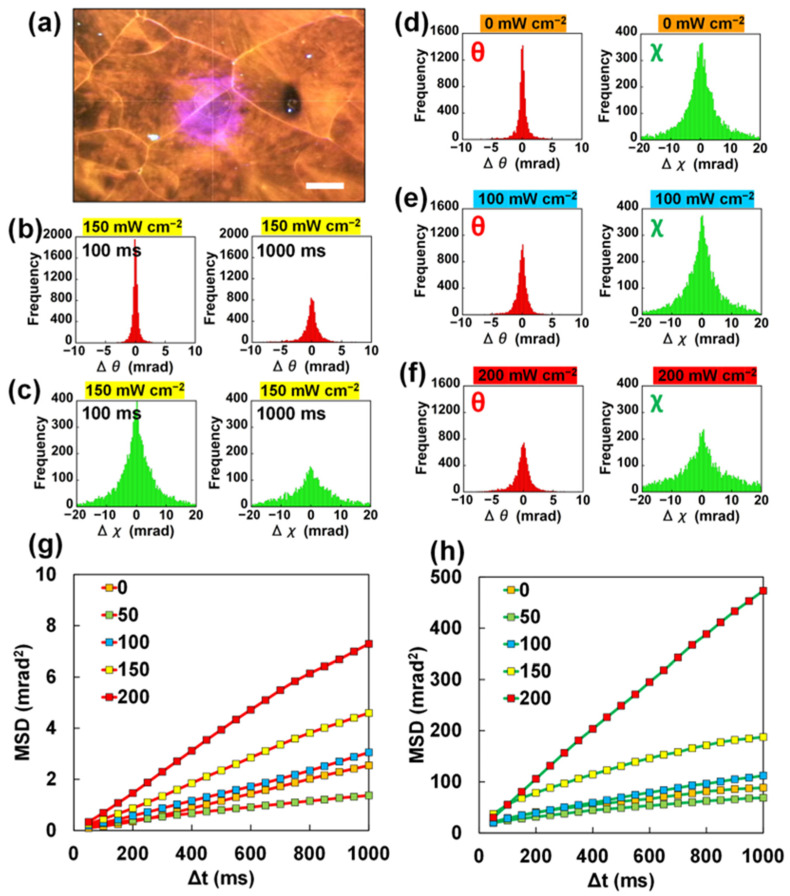
(**a**) Photograph of a 4-MAAB crystalline film containing ZnO particles (scale bar: 200 μm); (**b**,**c**) histograms of rotational displacements of ZnO particles in a 4-MAAB crystalline film in (**b**) θ direction and (**c**) χ direction at the light intensity of 150 mW cm^−2^. Displacements detected during the time interval (Δt) of 100 ms and 1000 ms are shown, respectively; (**d**–**f**) histograms of rotational displacements of ZnO particles in the 4-MAAB crystalline film in θ direction and χ direction at the light intensities (**d**) 0, (**e**) 100, and (**f**) 200 mW cm^−2^. Displacements detected during the time interval (Δt) of 1000 ms are shown; and (**g**,**h**) MSD curves of (**g**) θ direction and (**h**) χ direction calculated from the rotational displacements as a function of the time interval (Δt) at the light intensities 0, 50, 100, 150, and 200 mW cm^−2^.

**Figure 4 ijms-24-17462-f004:**
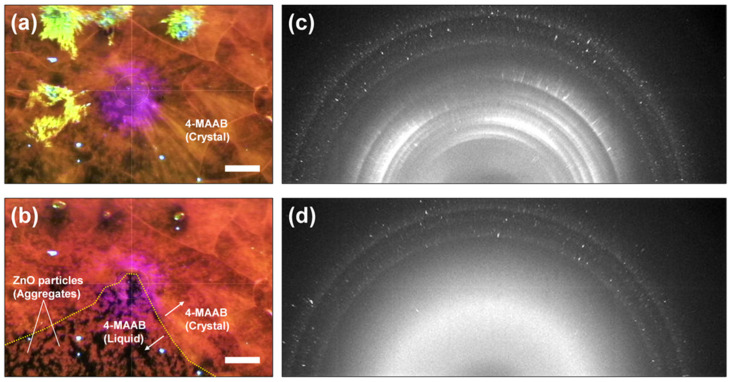
(**a**,**b**) Photographs of a 4-MAAB crystalline film containing ZnO particles before (**a**) and after (**b**) irradiation with 200 mW light (scale bars: 200 μm); (**c**,**d**) XRD patterns before (**c**) and after (**b**) light irradiation. XRD measurements were performed at the center of the images in (**a**,**b**).

**Figure 5 ijms-24-17462-f005:**
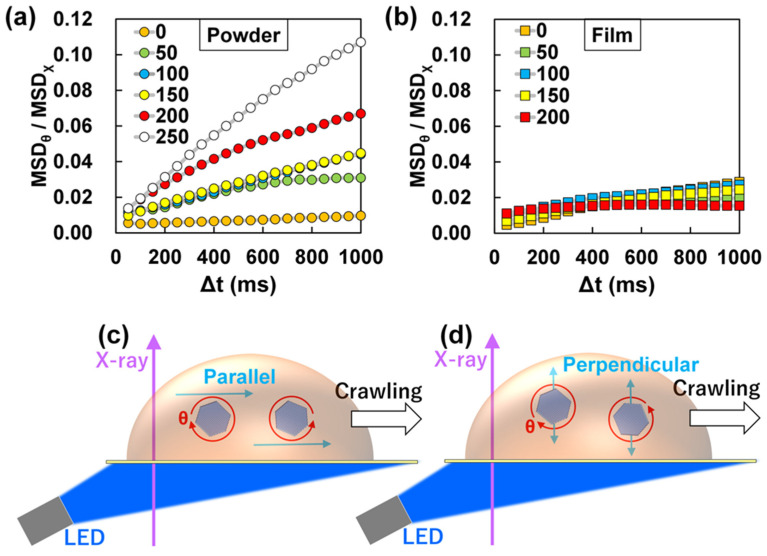
(**a**,**b**) MSD_θ_/MSD_χ_ curves of ZnO particles (**a**) for the powder and (**b**) for the film of 4-MAAB. The MSD_θ_/MSDχ curves at each irradiation light intensity are shown; and (**c**,**d**) schematic illustration of the rotation of ZnO particles in a 4-MAAB crystal driven by flow in directions (**c**) parallel and (**d**) perpendicular to the direction of the crawling motion.

## Data Availability

Data are contained within the article and Appendix A.

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
