# Peer review of "Visualization of the Dynamics of Photoinduced Crawling Motion of 4-(Methylamino)Azobenzene Crystals via Diffracted X-ray Tracking"

_ijms, 2023, doi:10.3390/ijms242417462_

Round 1
Reviewer 1 Report
Comments and Suggestions for Authors
The authors in this paper used a time-resolve X-ray diffraction method to track ZnO nanoparticle behaviors during the movements of the azobenzene (4-MAAB) crystals, which undergo reversible trans-to-cis photoisomerization accompanied with a crystal-to-liquid phase transition process. The authors also revealed that the ZnO nanoparticles would rotate more violently when light intensity increased. Moreover, the crawling motions of the 4-MAAB would initiate anisotropic rotation of the ZnO nanoparticles. The results presented in this paper provide a new strategy to illustrate the photoinduced crystal-to-liquid transition in photoresponsive crystals and might be helpful to the community of responsive crystals, crystal engineering, and functional materials. The paper is suggested to be accepted by the International Journal of Molecular Sciences (IJMS) after some revision. The questions and comments are as follows:
1. The authors are suggested to incorporate molecular structure and photochemical reaction scheme in the manuscript to illustrate the photochemistry process during the process.
2. It seems that the authors avoid discussing the velocity of the sample (azobenzene) during the moving proves.
3. I wonder why the authors used ZnO nanoparticles. Based on the SEM image of the ZnO nanoparticles, the size is not very uniform, and I wonder if the wide variation of ZnO particle size will influence the movements of 4-MMAB.
4. It is relatively difficult to track ZnO particles and 4-MMAB in the photographs, especially in Figure 4a and Figure 4b. The authors are suggested to add some markers or signs to the photographs to show ZnO and 4-MAAB.

Author Response
Comment 1. The authors are suggested to incorporate molecular structure and photochemical reaction scheme in the manuscript to illustrate the photochemistry process during the process.
Reply. The molecular structure and photoisomerization scheme of 4-MAAB have been added as Figure 1a.
Comment 2. It seems that the authors avoid discussing the velocity of the sample (azobenzene) during the moving proves.
Reply. Thank you for the insightful comment. Because the DXT experimental setup did not allow us to observe the 4-MAAB crystals at high magnification, it was difficult to clearly observe the crawling motion. Therefore, it was impossible to discuss the velocity of the crawling motion during the DXT experiment. However, the significant discovery in this study is that the photoinduced crawling motion causes anisotropy in the rotational direction of the ZnO particles in 4-MAAB. The relation between the velocity of crawling motion and the particle rotation is a topic for future study.
On the other hand, we show the photoinduced crawling motion of 4-MAAB crystals containing ZnO nanoparticles on a polyimide film in Figure S4. Although we cannot discuss the velocity in detail, we calculated the approximate average velocities of the 4-MAAB crystals from Figure S4 as supplementary information. We have added it to lines 304-306 in the manuscript.
Comment 3. I wonder why the authors used ZnO nanoparticles. Based on the SEM image of the ZnO nanoparticles, the size is not very uniform, and I wonder if the wide variation of ZnO particle size will influence the movements of 4-MMAB.
Reply. In our previous report, we have shown that silica particles with a diameter of 3 μm can be transported using a 4-MAAB crystal. Therefore, submicron-sized ZnO nanoparticles used here can be transported regardless of size variations. We have added the explanation to lines 116-119 in the manuscript.
Comment 4. It is relatively difficult to track ZnO particles and 4-MMAB in the photographs, especially in Figure 4a and Figure 4b. The authors are suggested to add some markers or signs to the photographs to show ZnO and 4-MAAB.
Reply. Thank you for the constructive suggestion. We have revised Figure 4a, b.
Reviewer 2 Report
Comments and Suggestions for Authors
A few minor remarks:
1)A typo in line 360 - "mannar".
2)The use of term "nanoparticles" for ZnO used it this work is slightly questionable, since "nanoparticles" are usually between 1 nm and 100 nm in size, while Figure S1 shows well-defined crystals up to 500 nm in size. Perhaps the term "fine particles" would be better in this case.
3)Similarly, 30 um thickness is rather large for "thin film". "polycrystalline film" would fit better in this case.
4)The term "powder crystal" also does not sound right. "Polycrystalline powder" or simply "4-MAAB powder" would be better.
Author Response
Comment 1. A typo in line 360 - "mannar".
Reply. Thank you. We have revised the manuscript.
Comment 2. The use of term "nanoparticles" for ZnO used it this work is slightly questionable, since "nanoparticles" are usually between 1 nm and 100 nm in size, while Figure S1 shows well-defined crystals up to 500 nm in size. Perhaps the term "fine particles" would be better in this case.
Reply. We have rewritten “ZnO nanoparticles” to “ZnO particles” or “fine particles”.
Comment 3. Similarly, 30 um thickness is rather large for "thin film". "polycrystalline film" would fit better in this case.
Reply. We have rewritten “thin film” to “film” or “polycrystalline film”.
Comment 4.The term "powder crystal" also does not sound right. "Polycrystalline powder" or simply "4-MAAB powder" would be better.
Reply. We have rewritten “powder crystal” to “powder” or “4-MAAB powder”.
Reviewer 3 Report
Comments and Suggestions for Authors
Manuscript IJMS-2719758 presents well documented and illustrated
experimental data allowing to visualize the dynamics of
photoinduced crawling motion of azobenzene microcrystals with embedded
zinc oxide nanoparticles.
I am sure that this article will be of interest to a wide range of readers,
so I recommend accept it for publication as it is.
Author Response
Thank you for the high evaluation of our research.
Reviewer 4 Report
Comments and Suggestions for Authors
The manuscript by Saito et al. discusses the very interesting topic of "crystal" motion under blue light irradiation. The main results (i.e., those concerning the motion) were obtained by diffracted X-ray tracking. The experiments are meaningful and designed with care and the data evaluation was well done.
There are however, some weaknesses concerning the precise description and the correct use of technical terms.
I made several annotations in the pdf file which I think should be considered before re-submission. Some points are listed in the following:
- “Nanoparticles” are, by definition, particles with at least one spatial dimension < 100 nm. Fig. S1 and the determined hydrodynamic radius speak a different language: The ZnO particles are no nanoparticles but just ZnO crystals.
- ZnO crystals in a solid matrix of polycrystalline 4-MAAB is not “a crystal”. It is a two-phase mixture or two-phase polycrystalline sample. The same holds for ZnO crystals in a polycrystalline 4-MAAB-film!
- The term “powder crystal” (line 140) does not exist. Most likely, the authors mean a polycrystalline sample.
- The “frequency” in the histograms is not explained. The number of “movements” of a reflection? An explanation is missing why the frequency decreases with increasing time and light intensity.
- I can hardly deduce any meaningful information from the photographs of the ZnO/4-MAAB mixtures (Fig. 3a, Fig. 4 a/b). Either a more detailed explanation (where precisely are the ZnO particles, e.g.) has to be added or the Figures can be removed.

Comments on the Quality of English LanguageAuthor Response
Comment 1. I made several annotations in the pdf file which I think should be considered before re-submission. Some points are listed in the following:
Reply. Thank you very much for checking the manuscript carefully. We have revised the manuscript according to the comments in the PDF. The replies to the questions in the PDF are listed below.
Comment 2. How does a "conventional" PXRD experiment of ZnO particles in molten 4-MAAB at ~ 90 °C, e.g., look like in a cylindrical sample shape, i.e. in a tube or capillary for comparison?
Reply. Because “conventional” PXRD uses monochromatic X-rays, it is impossible to track the movement of the diffraction spots from ZnO particles. Therefore, we did not conduct conventional PXRD experiments. However, the diffraction spots may blink on the diffraction ring from ZnO. An analysis method for particle dynamics that uses such blinking is called diffracted X-ray blinking (DXB). Sasaki et al. have reported on the method (Sci. Rep. 2018, 28, 17090).
Comment 3. Did the authors by any chance measure the UV/Vis spectrum of 4-MAAB? Especially in the range of the LED blue emission?
Reply. We have shown the absorbance spectrum of 4-MAAB in our previous paper (Adv. Mater. Interf., 2023, 10, 2202525.). Please see this paper published in an open access journal.
Comment 4. “Nanoparticles” are, by definition, particles with at least one spatial dimension < 100 nm. Fig. S1 and the determined hydrodynamic radius speak a different language: The ZnO particles are no nanoparticles but just ZnO crystals.
Reply. We have rewritten “ZnO nanoparticles” to “ZnO particles” or “fine particles”.
Comment 5. ZnO crystals in a solid matrix of polycrystalline 4-MAAB is not “a crystal”. It is a two-phase mixture or two-phase polycrystalline sample. The same holds for ZnO crystals in a polycrystalline 4-MAAB-film!
Reply. Thank you for the comments based on reviewer’s expertise. In this paper, in order to simplify the expression, the sample was referred to as "4-MAAB crystal". Following reviewer’s comment, we have added the explanation to lines 111-113 that the sample is “a two-phase polycrystalline sample” to be precise.
Comment 6. The term “powder crystal” (line 140) does not exist. Most likely, the authors mean a polycrystalline sample.
Reply. We have rewritten “powder crystal” to “powder” or “4-MAAB powder”.
Comment 7. The “frequency” in the histograms is not explained. The number of “movements” of a reflection? An explanation is missing why the frequency decreases with increasing time and light intensity.
Reply. The frequency in the histogram is the number of diffraction spots that appear in the XRD pattern. The central bin of the histogram represents zero displacement, that is, the number of diffraction spots that did not move during light irradiation. Therefore, the decrease in the central bin means the decrease in the number of ZnO particles that did not rotate.The rotational displacement of ZnO particles was calculated from the moving distance of the diffraction spot. As shown in Figure 1c, the rotational displacement of ZnO particles exhibits positive or negative values. Therefore, the increase in bins left and right from the center of the histogram means the increase in rotated ZnO particles. Following reviewer’s comment, we have added the description to lines 162-170.
Comment 8. I can hardly deduce any meaningful information from the photographs of the ZnO/4-MAAB mixtures (Fig. 3a, Fig. 4 a/b). Either a more detailed explanation (where precisely are the ZnO particles, e.g.) has to be added or the Figures can be removed.
Reply. Thank you for the constructive comment. Figure 3a is necessary to demonstrate that a 4-MAAB crystalline film was used in the DXT experiment. Without this image, it would be difficult to see the difference from Figure 2, which uses 4-MAAB powder. Figure 4a, b are necessary to demonstrate the photoinduced phase transition from crystal to liquid. Following the reviewer’s comment, we have revised Figure 4a, b to make it easier to understand.
Reviewer 5 Report
Comments and Suggestions for Authors
This paper continues a long-term study of the motion of azobenzene crystals during irradiation by light. The novel finding reported is that the use of Diffracted X-ray Tracking (DXT) on embedded ZnO nanoparticles provides details of rotational motions occurring inside the crystals as they partially melt, recrystallize, and crawl under the influence of illumination.
The methodology appears sound and is well-reported, and the conclusions are in line with the experimental results. A few questions come to mind:
1. Around line 201, the experiment on a crystalline film of 4-MAAB is reported to be done at 75 degrees. What temperature was used for the powder sample? Could temperature affect the results?
2. The caption for Figure S3 is very brief. What illumination was used?
3. In Figures 1 and 5, it looks as if the supporting film is horizontal and the X-ray beam is vertical. That obviously cannot be the case if the beam is from a storage ring. Correspondingly, the use of "vertical" (lines 38, 319-322, 325) to mean "perpendicular to the film" seems inappropriate. If the film is actually vertical, would gravity have an effect?
Comments on the Quality of English LanguageThe English is very good. In just a few places I think the wording could be better:
Line 52: "irradiating monochromatic" should be "irradiating with monochromatic".
Line 86-87: "became more intense depending on the irradiation light intensity" would be better as "increased as the intensity of the irradiating light increased".
line 99: "spots corresponds to 4-MAAB crystal" would be better as "spots arise from the 4-MAAB powder".
Line 127: "X-ray" should be "X-rays".
Line 237: "liquefied and a dark background was visible" would be better as "liquefied appeared dark due to the background showing through".
Line 259: "increasing mannar" should be "manner of increase".
Lines 261-262: "exponential increase manner" should be "exponential increase in MSD", and "saturated increase manner" should be "saturating increase in MSD".
Line 283: "inside" should be removed.
Line 299: "is originated" should be "originates".
Line 316: "improvement" would be better as "enhancement".
Line 353: "As a result," would be better as "This process produced".
Author Response
Comment 1. Around line 201, the experiment on a crystalline film of 4-MAAB is reported to be done at 75 degrees. What temperature was used for the powder sample? Could temperature affect the results?
Reply. The DXT experiment on the 4-MAAB powder sample were carried out under room temperature (25 °C). We have added the experimental condition to line 129 in the manuscript. The increase in temperature of the sample promotes photomelting of the 4-MAAB crystals. Here, figure 3 g,h indicates that the rotational motion of the ZnO particles increased due to the photomelting. However, as shown in Figure 5, the rotational anisotropy does not depend on the photomelting. Therefore, it is suggested that temperature does not have a significant effect on the rotational anisotropy.
Comment 2. The caption for Figure S3 is very brief. What illumination was used?
Reply. The experimental conditions are the same as in Figure 3g, h. Following the reviewer's comments, we have added the illumination condition to the caption for Figure S3.
Comment 3. In Figures 1 and 5, it looks as if the supporting film is horizontal and the X-ray beam is vertical. That obviously cannot be the case if the beam is from a storage ring. Correspondingly, the use of "vertical" (lines 38, 319-322, 325) to mean "perpendicular to the film" seems inappropriate. If the film is actually vertical, would gravity have an effect?
Reply. Thank you for pointing out our misunderstanding. As the reviewer commented, the supporting film of the 4-MAAB crystals was "vertical" for the horizontal X-ray. Therefore, it was inaccurate to refer to the direction perpendicular to the support film as "Vertical." We have revised the relevant text (lines et al. 37, 335-340) and figure 5d. Due to the small size of the 4-MAAB crystals, it is assumed that gravity has little effect on photoinduced crystal migration. In fact, in a previous report (AMI), we observed 4-MAAB crystals moving in a direction that defies gravity. We have added the explanation to line 120-123 and 126-128 in the manuscript.
Comments on the Quality of English Language
The English is very good. In just a few places I think the wording could be better:
Line 52: "irradiating monochromatic" should be "irradiating with monochromatic".
Line 86-87: "became more intense depending on the irradiation light intensity" would be better as "increased as the intensity of the irradiating light increased".
line 99: "spots corresponds to 4-MAAB crystal" would be better as "spots arise from the 4-MAAB powder".
Line 127: "X-ray" should be "X-rays".
Line 237: "liquefied and a dark background was visible" would be better as "liquefied appeared dark due to the background showing through".
Line 259: "increasing mannar" should be "manner of increase".
Lines 261-262: "exponential increase manner" should be "exponential increase in MSD", and "saturated increase manner" should be "saturating increase in MSD".
Line 283: "inside" should be removed.
Line 299: "is originated" should be "originates".
Line 316: "improvement" would be better as "enhancement".
Line 353: "As a result," would be better as "This process produced".
Reply. Thank you very much for checking the manuscript carefully. We have revised the manuscript according to the comments.